# Primate amygdala neurons evaluate the progress of self-defined economic choice sequences

**Fabian Grabenhorst[1]\*[†], Istvan Hernadi[1,2,3][†], Wolfram Schultz[1]**

[1]Department of Physiology, Development and Neuroscience, University of Cambridge, Cambridge, United Kingdom; [2]Department of Experimental Zoology and Neurobiology, University of Pécs, Pécs, Hungary; [3]Grastyan Translational Research Centre, University of Pécs, Pécs, Hungary

**Abstract** The amygdala is a prime valuation structure yet its functions in advanced behaviors are poorly understood. We tested whether individual amygdala neurons encode a critical requirement for goal-directed behavior: the evaluation of progress during sequential choices. As monkeys progressed through choice sequences toward rewards, amygdala neurons showed phasic, gradually increasing responses over successive choice steps. These responses occurred in the absence of external progress cues or motor preplanning. They were often specific to self-defined sequences, typically disappearing during instructed control sequences with similar reward expectation. Their build-up rate reflected prospectively the forthcoming choice sequence, suggesting adaptation to an internal plan. Population decoding demonstrated a high-accuracy progress code. These findings indicate that amygdala neurons evaluate the progress of planned, self-defined behavioral sequences. Such progress signals seem essential for aligning stepwise choices with internal plans. Their presence in amygdala neurons may inform understanding of human conditions with amygdala dysfunction and deregulated reward pursuit.

\*For correspondence: fg292@ cam.ac.uk

[†]These authors contributed equally to this work

## Introduction

The amygdala, a nuclear complex in the medial temporal lobe, is a key structure for the internal evaluation of sensory events (*Janak and Tye, 2015*; *LeDoux, 2000*; *Morrison and Salzman, 2010*; *Murray and Rudebeck, 2013*; *Rolls, 2000*). In addition to the classical role in fear conditioning (*Duvarci and Pare, 2014*; *LeDoux, 2000*; *Namburi et al., 2015*; *Paz and Pare, 2013*), amygdala neurons also respond to reward, encode stimulus-reward associations and signal reward expectation (*Belova et al., 2007*; *Bermudez and Schultz, 2010*; *Bermudez et al., 2012*; *Namburi et al., 2015*; *Nishijo et al., 1988*; *Paton et al., 2006*; *Rolls, 2000*; *Schoenbaum et al., 1998*; *Schultz, 2015*; *Shabel and Janak, 2009*). A central, integrative valuation role is compatible with the amygdala's extensive sensory and prefrontal connections, its outputs to attentional, autonomic, and behavioral control systems (*Duvarci and Pare, 2014*; *LeDoux, 2000*; *Price, 2003*), and its processing of social cues (*Adolphs, 2010*; *Gothard et al., 2007*; *Leonard et al., 1985*; *Livneh et al., 2012*; *Mosher et al., 2014*; *Rutishauser et al., 2015*). Accordingly, amygdala lesions across different species impair reward-related behaviors (*Balleine and Killcross, 2006*; *Baxter and Murray, 2002*; *Baylis and Gaffan, 1991*; *Bechara et al., 1999*; *Brand et al., 2007*; *Everitt et al., 2003*; *Rudebeck et al., 2013*). However, beyond basic reward processing, the functions of amygdala neurons in advanced behaviors, such as economic decision-making, are poorly understood.

Advanced economic behaviors often require sequences of choices organized by an internal plan. In economic saving, for example, the plan to obtain a reward in the future guides sequential choices

and actions in the present (*Benhabib and Bisin, 2005*). A critical requirement during planned choice sequences is the continual evaluation of progress, to ensure alignment of individual choices with long-term plans and successful navigation toward reward goals (*Benhabib and Bisin, 2005*; *Berkman and Lieberman, 2009*; *Johnson and Busemeyer, 2001*). Physiological studies of frontal cortex identified activity patterns underlying action sequences and the execution and updating of motor plans (*Mushiake et al., 2006*; *Shima and Tanji, 1998*; *Shima et al., 1996*; *Sohn and Lee, 2007*; *Tanji and Shima, 1994*). Furthermore, during externally instructed, multistep reward schedules, neurons in frontal cortex, amygdala and striatum respond to cues signaling schedule onset, reward contingency and reward proximity (*Shidara and Richmond, 2002*; *Simmons et al., 2007*; *Sugase-Miyamoto and Richmond, 2005*). These studies uncovered important neural components of externally instructed sequential behaviors but did not address the internal progress evaluation that is typical of economic choice plans toward self-defined goals.

Here we tested whether amygdala neurons encode the internally evaluated progress of economic choice sequences. The amygdala is a prime candidate structure for this process. Its diverse anatomical connections would make a sequence progress signal widely available for cognitive, motivational and behavioral computation in different downstream neurons. Moreover, amygdala dysfunction occurs in humans with deregulated reward pursuit and affective disorders (*Koob and Volkow, 2010*; *Price and Drevets, 2010*), which impact on the motivation to plan for and pursue distant rewards. In a series of experiments (*Grabenhorst et al., 2012*; *Hernadi et al., 2015*), we recorded amygdala single-neuron activity while monkeys performed self-defined choice sequences toward distant rewards. Previously, we reported prospective amygdala signaling of immediate, current-trial choices (*Grabenhorst et al., 2012*) and forthcoming choice sequences (*Hernadi et al., 2015*). Although such signals are essential for internally planned behavior, they cannot by themselves guide stepwise choices toward a final reward goal.

In the present study, we report gradually increasing amygdala responses that evolved dynamically during internally planned choice sequences lasting up to two minutes. These gradual responses had several remarkable characteristics. They occurred in the absence of external progress cues that could have induced sensory evidence accumulation and in the absence of opportunities for motor preplanning. The gradual amygdala responses were mostly specific to free, internally guided choices as they typically disappeared in externally instructed sequences, despite similar behavioral reward expectation. Importantly, their gradual build-up rate readily adapted to the forthcoming, internally planned choice sequence length, suggesting reference to an internal plan. Decoding analysis showed that the neuronal population provided a highly accurate progress code. These data suggest that primate amygdala neurons encode internally evaluated progress during planned economic choice sequences. Such gradual activity seems suited to support multistep decision-making and motivational regulation during the pursuit of self-defined distant rewards.

## Results

### Sequential choice task and economic behavior

Two monkeys performed in a sequential choice task in which they made consecutive, trial-by-trial choices to save (i.e. accumulate) liquid reward until they chose to spend (i.e. consume) the accumulated reward (*Figure 1A*). The animals indicated their save-spend choice on each trial by a saccade towards the save or spend cue. Accumulation of reward over consecutive save choices depended on the current interest rate; pre-trained save cues indicated different interest rates. The animals were free to produce saving sequences of different lengths, which allowed them to plan their behavior over several trials within a sequence and anticipate final reward amounts. Save-spend cues appeared pseudorandomly in left-right positions, preventing action planning before cue onset. To compare internally guided choices with externally instructed behavior, we tested 'imperative' save-spend sequences of comparable lengths.

A critical task parameter for the animals was the number of completed trials since the start of the current saving choice sequence. This parameter, which we term 'sequence progress', evolved dynamically within a sequence and thus determined both reward proximity and accumulated reward amount (*Equation 1*). Crucially, neither free choice task nor imperative control task provided

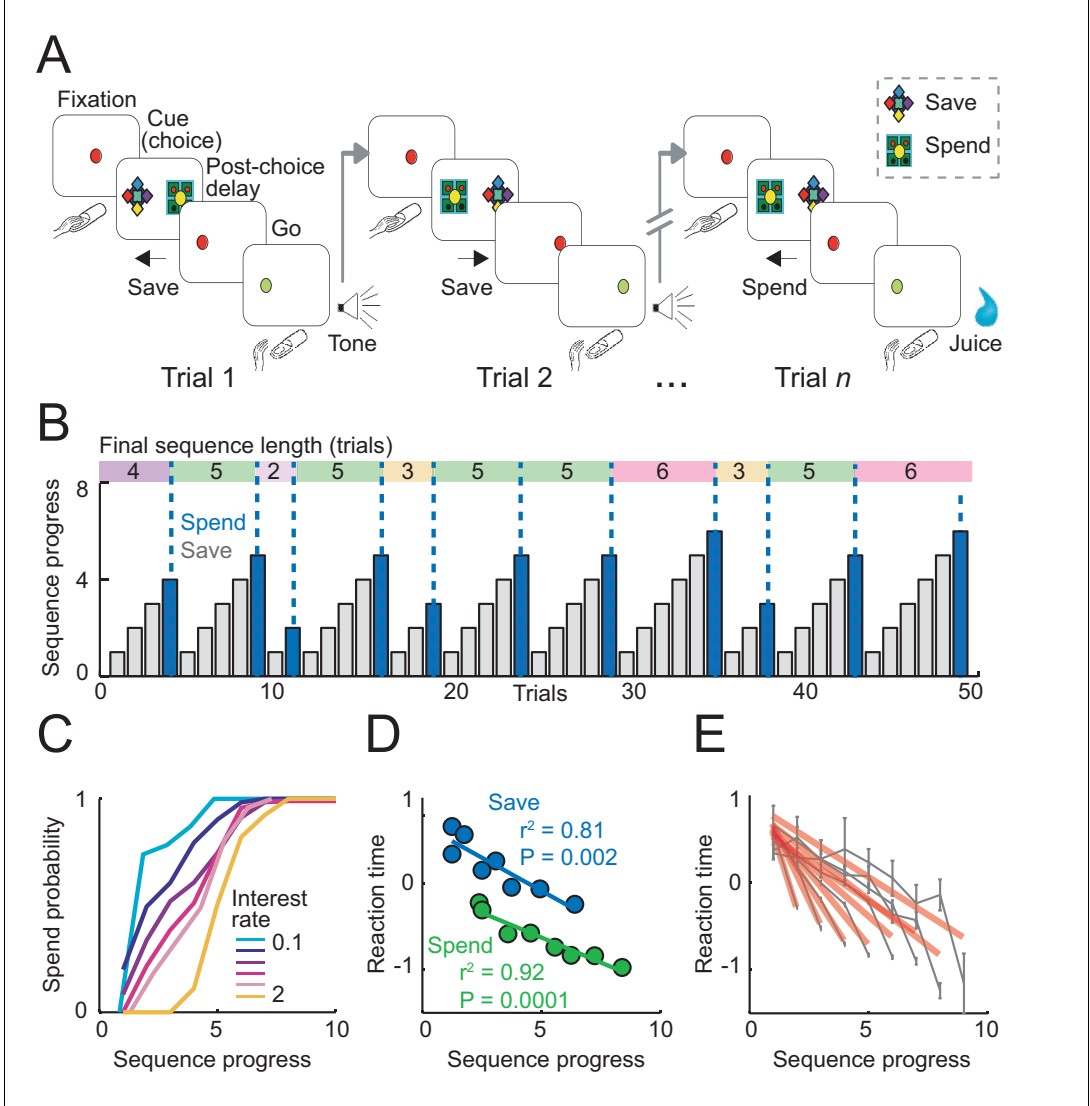

**Figure 1.** Sequential choice task and behavior. (**A**) Monkeys made sequential choices to save (accumulate) or spend (consume) reward. Consecutive save choices increased reward amount according to the current interest rate; spend choice resulted in reward delivery. Different save cues indicated interest rates. Choice sequences lasted up to nine consecutive trials (~12 s per trial) and were self-determined, which allowed the animals to plan their behavior several steps in advance. (**B**) Example behavioral data. Bars show trial-by-trial choice record (gray: save, blue: spend). Sequence progress corresponds to the cumulative trial record within each sequence. Colored boxes delineate saving sequences; numbers indicate the sequence length. (**C**) Increases in spend choice probability as a function of sequence progress, shown for different interest rate conditions. (**D**) Reaction times decreased with sequence progress on both spend and save trials (linear regression; mean reaction times for equally populated sequence progress bins; error bars smaller than symbols). Reaction times in this figure were measured as the latencies with which the animals released a touch key at the end of a trial to initiate reinforcer delivery (liquid reward on spend trials, auditory cue on save trials). (**E**) Reaction times adapted to sequence length. Gray data show reaction times as a function of sequence progress separately for different sequence lengths (N = 19,612 trials). Red lines show fits from linear regression.

The following figure supplement is available for figure 1:

**Figure supplement 1.** Reaction times analyses.

external cues about how many trials had been completed within a sequence, requiring the animals to track progress internally.

In a typical session, the animals produced saving sequences of different lengths (*Figure 1B*). The behavioral relevance of sequence progress can be assessed by its relationship with the animals'

choice probability. By design, spend choice probability increased with sequence progress (as each sequence ended with a spend choice). However, the relationship between spend probability and sequence progress varied between interest rate conditions, reflecting the animals' preferences for different sequence lengths (*Figure 1C*). The animals typically preferred shorter sequences when interest was low (early peak in the probability distribution) and preferred longer sequences with higher interest (later peak; correlation between mean sequence length and interest rates across animals: r = 0.71, p=0.003). Thus, the animals' choices depended on sequence progress.

Analysis of reaction times confirmed the behavioral relevance of sequence progress. We focused our main reaction time analysis on the latencies with which the animals released a touch key at the end of a trial to initiate reinforcer delivery (liquid reward on spend trials, auditory cue on save trials). This measure has previously been shown to reflect a differential reward expectation related to reward type (*Watanabe et al., 2001*) and reward magnitude (*Hernadi et al., 2015*), with typically faster responses for preferred rewards and high magnitude rewards. We focused on key release latencies because in the present task they may reflect more closely the animal's reward expectancy compared to saccadic reaction times, as key release was the final behavioral response required before reinforcer delivery. Key release reaction times shortened with sequence progress, separately for spend and save trials (*Figure 1D*), which suggested higher motivation in later trials of a sequence. The effect was significant when controlling for other task-relevant variables including subjective choice values (p<0.001, partial correlation). The relationship between reaction times and sequence progress depended on the final length of each sequence: reaction times declined at a faster rate with the progress of shorter sequences (*Figure 1E*), indicating that the animals adjusted their motivation according to planned sequence length. In the imperative control task, reaction times also decreased with sequence progress depending on anticipated sequence length (*Figure 1—figure supplement 1*). Statistical modeling of reaction times from both tasks in the same multiple regression model suggested that the strongest factors across tasks were sequence progress and current-trial save-spend choice. An additional regressor for task type (free choice vs. imperative task) and its interaction with other variables accounted for insignificant portions of variance (*Figure 1—figure supplement 1C*). Critically, the influence of sequence progress on reaction times did not depend on task type, suggesting similar behavioral relevance for sequence progress across tasks. Saccadic reaction times, measured as the latency from the choice cue onset to fixation of the chosen cue were also related to sequence progress (*Figure 1—figure supplement 1D*), although the effect was smaller compared to key release latencies and had a positive rather than negative direction. (We are cautious in interpreting saccadic reaction times, as touch key reactions were overall more directly related to the animals' motivation: release of touch key was the final behavioral response on each trial before reinforcer delivery and, in further analyses, touch key release latencies showed a stronger relationship to upcoming reward magnitude on spend trials ($R^2$ = 0.116, p<0.0001) compared to saccade reaction times ($R^2$ = 0.017, p<0.001). Overall, reaction time analyses confirmed that sequence progress was a behaviorally and motivationally important parameter.

Behavioral data from control tests reported previously (*Hernadi et al., 2015*) confirmed that saving was adaptive and internally controlled: both animals (i) followed uncued changes in interest rate (R = 0.97, p<0.001; linear regression of mean sequence length on interest rate); (ii) maximized reward rate for extreme interest conditions (R = 0.73, p=3.3 $\times$ $10^{-7}$, linear regression of relative choice frequency on normalized rate of reward return across animals); and (iii) successfully tracked accumulated reward, shown by interspersed challenge trials offering saved vs. fixed reward amounts (p=5.06 $\times$ $10^{-6}$, Mann-Whitney test on choice frequencies for fixed reward when it exceeded saved reward). In addition, the animals' choices were well described by a behavioral model based on the subjective values of save and spend choice options (p<1.0 $\times$ $10^{-16}$, t-test on regression coefficients for subjective save value and spend value, logistic regression). Taken together, the animals' saving choice sequences were economically meaningful and internally planned. Sequence progress was a key task parameter that influenced the animals' choices and motivation.

## Encoding of sequence progress by individual amygdala neurons

As the animals progressed through the self-determined choice sequences, a substantial number of amygdala neurons showed gradually changing neuronal responses over consecutive trials in a sequence (single linear regression, *Equation 2*: 194/326 neurons, 59%; multiple stepwise regression, *Equation 7*: 127/326 neurons, 39%). The activity of these neurons thus seemed related to sequence

progress. We identified such neurons by regressing their trial-by-trial activity in fixed time windows on the current, cumulative number of trials in a sequence (p<0.05, linear regression, *Equation 2*, following selection of task-related responses: p<0.0083, Wilcoxon test, corrected for multiple comparisons). Additional sliding window and multiple regression analyses confirmed the main results and controlled for other behavioral and task variables, as detailed below.

The neuron in *Figure 2* showed a phasic response related to sequence progress at the time of choice on each trial. The activity increased gradually over consecutive trials within a sequence and reset with the start of a new sequence (*Figure 2A*). This gradual activity occurred specifically in the cue phase and preceding fixation phase (*Figure 2B*); it was neither a tonic increase in baseline activity nor a trial outcome response. Crucially, there were no external cues that could have signaled current trial position; accordingly, the observed activity reflected purely internal tracking of sequence

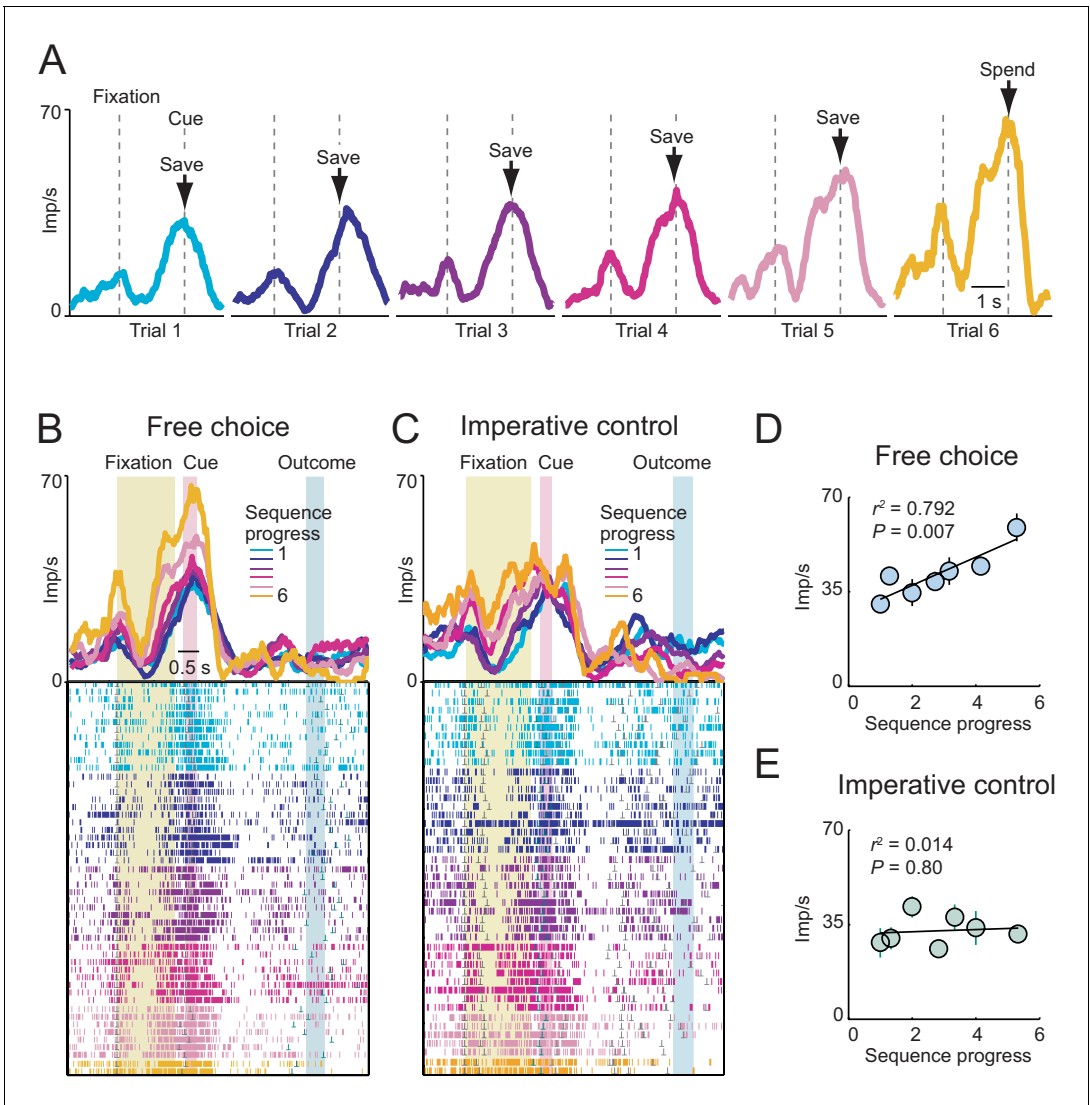

**Figure 2.** Gradually increasing activity related to sequence progress in a single amygdala neuron. (A) Neuronal response in the fixation and cue periods, plotted for each step of a six-trial choice sequence. Color indicates sequence progress. (B) Peri-event time histogram for the same neuron, aligned to fixation onset, sorted by sequence progress. Raster display: ticks indicate impulses, rows indicate trials; grey dots indicate event markers (labelled above graph). Activity in cue phase and preceding fixation phase reflected sequence progress, before the animal indicated its current-trial choice. Visual stimulation during fixation was constant over consecutive trials. Randomized cue positions precluded preplanning of action sequences. (C) The progress-related activity in the cue phase disappeared during performance of the imperative control task. (D, E) Linear regression (*Equation 2*) of trial-by-trial cue-phase activity on sequence progress in free choice and imperative task (error bars: s.e.m.).

progress. The graded activity disappeared in the cue phase of the imperative task, when behavior was externally instructed (*Figure 2C*), confirming the relationship to an internal process. Regression analysis (*Equation 2*) established a linear, trial-by-trial relationship between cue-period activity and sequence progress in the free choice task (*Figure 2D*) but not in the imperative task (*Figure 2E*). Importantly, sensory stimulation, motor requirements and behavioral reward expectation were matched between free choice and imperative task, and random cue positions precluded planning of action sequences. This suggested that the gradually increasing activity did not reflect sensory evidence accumulation or preparation of a motor response. Thus, at the time of choice on each trial, the neuron showed gradually increasing activity related to sequence progress, specifically during internally guided choice sequences.

Linear regression (*Equation 2*) identified amygdala neurons with progress-related activity in different trial periods (*Figure 3A,B*), including pre-choice periods before the animal indicated its current-trial choice (*Figure 3A*, left), and post-choice periods before outcome (*Figure 3A*, middle) and after outcome (*Figure 3A*, right). A significant majority of progress-related activities increased (rather than decreased) over consecutive trials within a sequence (*Figure 3C*; 294/408 responses with positive regression slope, *Equation 2*, 72%; p=1.0 $\times$ 10$^{-19}$, binomial test). We compared neuronal progress encoding between free choice and imperative tasks in a subset of neurons recorded in both tasks (N = 155). Although a fraction of these neurons also showed progress-related activity in the imperative task, the significant majority showed progress-related activity specifically during free choices (*Figure 3D*; p<0.001, z-test for dependent samples). This suggested that progress signals were mostly specific to a situation that required internally guided behavior. Consistent with this observation, at the population level, the strength of neuronal progress coding transiently declined on trials when the animals committed an error (failed to perform a correct saccade or prematurely released the touch key), and subsequently reappeared on the following trials (*Figure 3E*). This result suggested a relationship between amygdala progress signals and the animals' performance. Sliding window regression identified progress activities in a similar numbers of neurons as the main fixed-window regression (214/326 neurons, 65.64%, *Equation 2*) and confirmed their occurrence throughout different task periods (*Figure 3F*). The majority of sequence progress activities were phasic responses that occurred transiently at specific task events and subsequently disappeared (*Figure 3F*), rather than reflecting lasting, tonic modulations in activity. Additional multiple regression models confirmed the robustness of these results and distinguished sequence progress activities from closely related task-relevant variables (*Figure 3—figure supplement 1*). For example, few neuronal responses (27/408, 6.6%) reflected sequence progress in addition to previously described sequence length signals, suggesting distinct encoding of sequence progress and length. Our strictest regression model included sequence progress in addition to planning variables (sequence value, sequence length, *Equation 7*) and current-trial save-spend choice. This model identified 127 neurons with relation to sequence progress (39%) of which 31 neurons (10% of all recorded neurons) encoded sequence progress without relation to explicit planning variables or choices, suggesting a significant number of neurons with 'pure' progress activity. For comparison, 78 neurons (24%) neurons encoded 'pure' planning activities (i.e. not coding choice or progress variables) and 112 neurons (34%) encoded 'pure' choice activities (i.e. not coding planning or progress variables). Thus, amygdala progress neurons were in many cases distinct from previously described planning and choice neurons (*Grabenhorst et al., 2012*; *Hernadi et al., 2015*). Histological reconstruction located the recording sites in amygdala. Although neurons with progress-related activity occurred in both basolateral and centromedial amygdala regions (*Figure 3G*), they occurred significantly more frequently in the basolateral amygdala (*Figure 3H*).

Taken together, a substantial number of amygdala neurons had gradually changing activity as the animals progressed through a choice sequence. These activities typically had a positive progress slope, were not explained by alternative task-relevant variables or planning activities, were largely specific to free choices, and tracked local fluctuations in the animals' performance.

## Adaptation of sequence progress activity to planned sequence length

Choice sequence lengths generated by the animals varied considerably, even within single sessions. Under such conditions, a neuron signaling sequence progress should adapt its sensitivity to the currently relevant range of sequence steps, to maximize information with a limited dynamic coding range (*Fairhall et al., 2001*; *Laughlin, 1981*; *Schultz, 2015*). The neuron in *Figure 4A–B*, showed

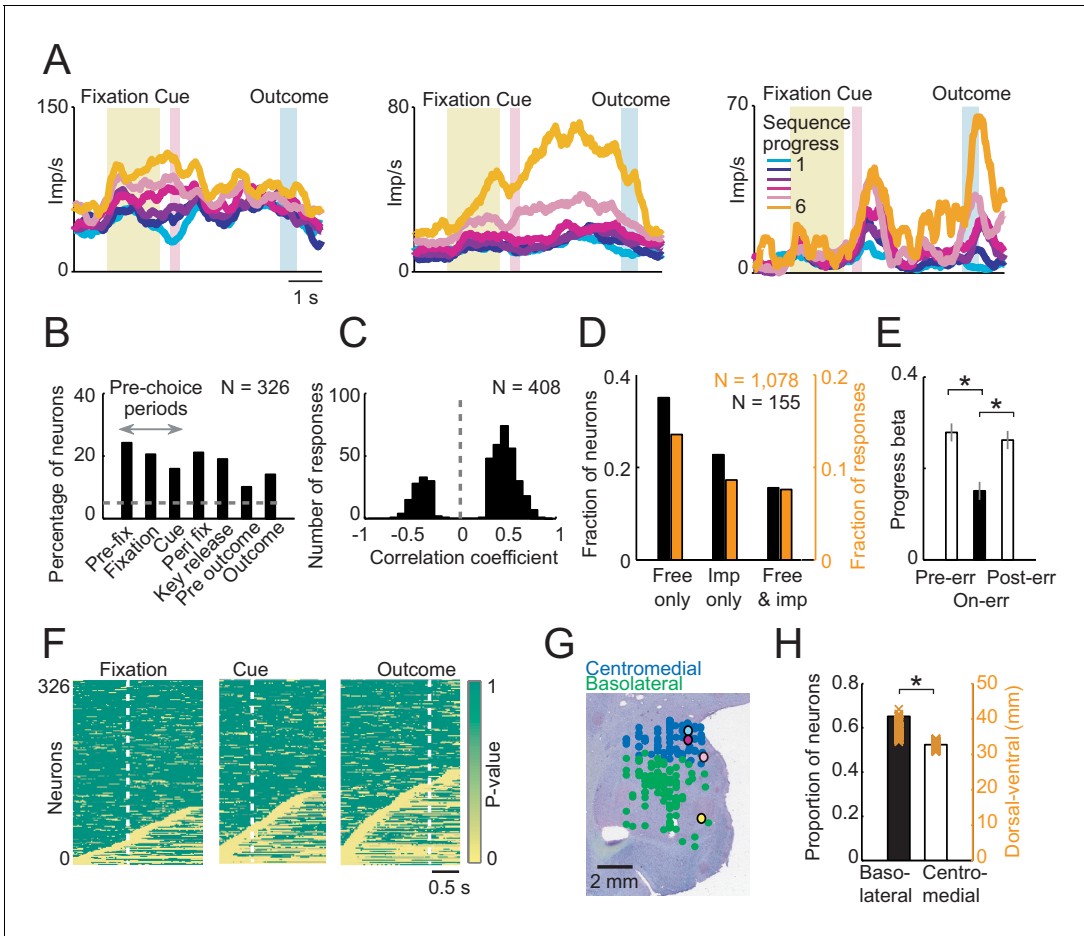

**Figure 3.** Sequence progress activity in the population of amygdala neurons. (**A**) Three single neurons with progress-related activity in fixation and cue phases (left), delay period from choice to outcome (middle) and outcome phase (right). (**B**) Percentages of neurons with progress-related activity (linear regression, *Equation 2*) for all task periods. (**C**) Distribution of regression slopes for all responses with significant sequence progress encoding (i.e. significant regression slopes aggregated across all task periods and neurons, *Equation 2*). (**D**) Comparison with imperative task. Fraction of neurons (black) and responses (orange) significant for sequence progress (*Equation 2*) in the free choice task only ('Free only'), imperative control task only ('Imp only') and both tasks ('Both'), based on 155 neurons (1078 responses) tested in both tasks. The fraction of neurons and responses significant in the free choice task only was significantly higher compared to both other categories ($p<0.001$, z-test for dependent samples). (**E**) Relationship to performance errors. Bars show standardized regression coefficients (± s.e.m) from a population analysis regressing normalized activity on sequence progress for trials preceding errors (Pre-err), error trials (On-err) and trials following errors (Post-err). The strength of progress coding was reduced on error trials compared to immediately preceding trials and subsequently reappeared following error trials ($p<0.001$; dependent-samples t-tests). (**F**) Statistical p-values for linear regression of activity on sequence progress in all neurons, obtained from sliding window analysis (*Equation 2*; window size: 200 ms, step size: 25 ms) aligned to fixation, cue and outcome events. Data in each row are from a single neuron, sorted from bottom to top within each panel according to coding latency. For clarity, p-values>0.05 were set to 1. (**G**) Histological reconstruction of recording sites for progress-sensitive neurons, shown separately for centromedial and basolateral amygdala regions. Yellow, pink, magenta, and cyan symbols: example neurons in *Figures 2* and *3A*. Collapsing across anterior-posterior dimension resulted in symbol overlap. (**H**) Proportion of neurons (N = 194) with sequence progress activity in basolateral and centromedial amygdala ($p=0.005$, $\chi^2$-test). Orange symbols: Recording depths of sequence progress neurons (dorsal-ventral axis, reference bregma).

The following figure supplement is available for figure 3:

**Figure supplement 1.** Percentages of progress-sensitive neurons obtained from different regression models.

exactly this kind of neuronal adaptation. During short sequences of less than five trials, progress activity rose steeply from baseline to a maximum of about 40 Hz (regression slope from unbinned data, *Equation 2*: 7.33), whereas for longer sequences, activity increased at a much lower rate (slope: 2.14). We further examined adaptation by pooling data across neurons. As for the single

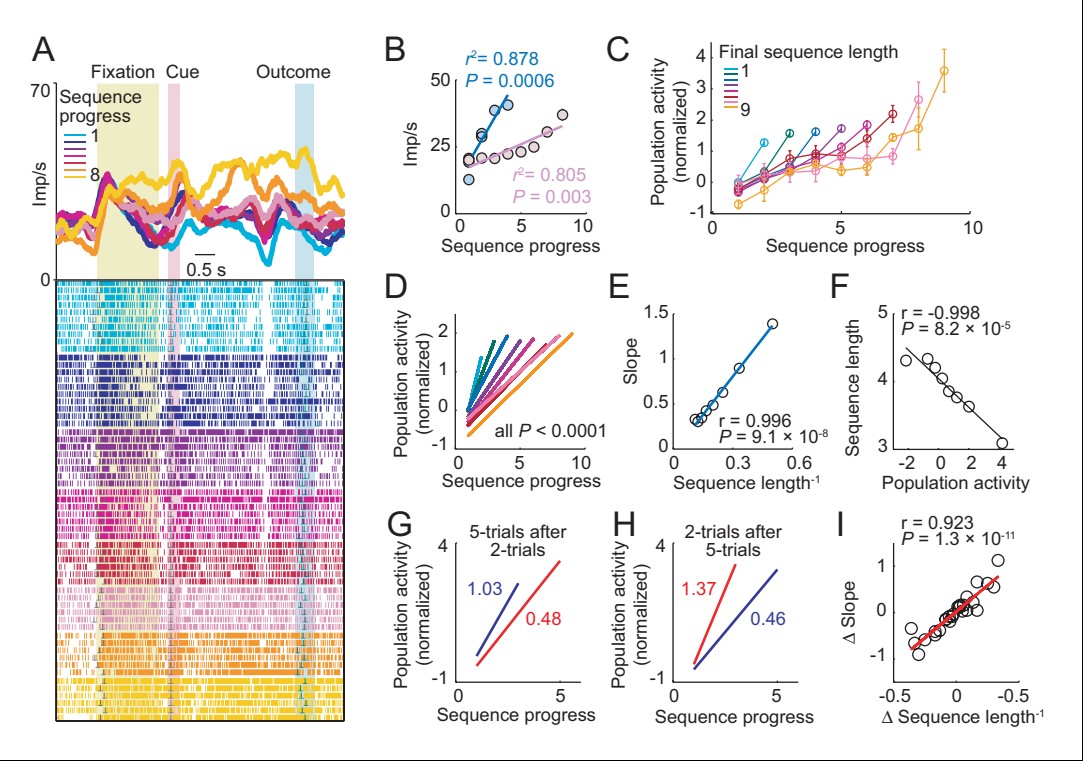

**Figure 4.** Amygdala progress activity adapts to internally planned sequence length. (A, B) Adaptation of progress activity in a single neuron. (A) The neuron showed progress-related activity in the delay period between cue and outcome events. (B) Linear regression (*Equation 2*) of the neuron's trial-by-trial delay-period activity on sequence progress for short (< five trials) and long (> four trials) sequences. Activity rose more steeply during short compared to long sequences (raw slopes: 7.33 vs. 2.14). Note that sequence lengths were internally planned rather than instructed, suggesting prospective adaptation to an internally planned parameter. (C) Adaptation to sequence length in population activity. Normalized activity for all neuronal responses significant for sequence progress (N = 408 responses from 194 neurons. *Equation 2*), plotted separately for different sequence lengths (mean ± s.e.m). (D) Linear regression fits to data in (C). (E) Slopes from linear fits in (D) plotted against the inverse sequence length (linear regression). (F) Prediction of final sequence length from population activity. Linear regression of sequence length on the population activity of progress-related neuron responses, measured on the second trial of each sequence. (G–I) Adaptation for consecutive choice sequences. (G) Linear fits of population activity to sequence progress for five-trial sequences and immediately preceding two-trial sequences. Colored numbers indicate slopes. (H) Linear fits for two-trial sequences and immediately preceding five-trial sequences. (I) Linear regression of slope differences (Δ Slope), obtained from neuronal activity during consecutive sequences, on corresponding differences in inverse lengths of consecutive sequences (Δ Sequence length$^{-1}$).

neuron, increases in population activity with sequence progress were steep for short sequences and successively shallower for longer sequences (*Figure 4C,D*). This was quantified by a linear positive relationship between the rate of activity increase (population progress slope) and inverse sequence length (*Figure 4E*). Thus, the rate of activity increase adapted to the length of the current choice sequence, consistent with principles of adaptive coding.

Crucially, choice sequence lengths were internally planned rather than externally instructed, which implied prospective neuronal adaptation to an internally planned parameter. Confirming a prospective adaptation, progress-related activity at specific sequence steps allowed prediction of the forthcoming sequence lengths, as early as the second trial of a sequence (*Figure 4F*). This effect was found for all sequence steps (all p<0.001, linear regression) except first sequence trials (p=0.26), which implied a constant starting point of progress activity in each sequence. Thus, the rate of activity increase with sequence progress depended on planned sequence length, which enabled prediction of sequence length at specific sequence steps.

We tested the temporal dynamics of neuronal adaptation by examining activity for pairs of consecutive sequences that differed in length. For example, consider a two-trial sequence followed by a five-trial sequence (*Figure 4G*). Accurate progress tracking would require a neuron to adapt its activity slope flexibly from one sequence to the next. This is exactly what we observed in population activity. After a steep activity increase during two-trial sequences, activity increase was much shallower during immediately following five-trial sequences (*Figure 4G*). This rapid downward adaption of progress slope reversed when sequential order was reversed, i.e. when a two-trial sequence followed a five-trial sequence (*Figure 4H*). We quantified this dynamic adaptation across all pairs of consecutive saving sequences by a significant regression of the (signed) progress activity slope difference on the (signed) difference in sequence length (*Figure 4I*). Thus, the amygdala progress activity adapted its rate of increase flexibly and rapidly from one sequence to the next to reflect the current planned sequence length.

## Adaptive model of sequence progress activity

The above findings suggest an adaptive code for sequence progress in amygdala neurons. Accordingly, we examined whether single-neuron activity was best fit by an adaptive model that normalized sequence progress by sequence length (*Equation 3*, *Figure 5A*). Linear regression revealed that this adaptive progress model explained a higher number of responses than a non-adaptive progress model (447 vs. 408 significant responses, *Equation 3* vs. *Equation 2*), with a significantly higher number of best fits (391 vs. 104 responses with higher $R^2$ for adaptive model; $p<1.0 \times 10^{-23}$, z-test for dependent samples). Across all responses that were significant for both models, the $R^2$ distribution was shifted significantly towards the adaptive model (*Figure 5B*, left) and the $R^2$ difference between models was significantly in favor of the adaptive model ($p=3.2 \times 10^{-8}$, one-sample t-test). A stepwise regression in which both models competed to explain neuronal responses also favored the adaptive model (284 vs. 121 significant responses). The adaptive sequence progress model also proved superior to other alternatives (*Figure 5A–B*), including models based on the accumulated reward magnitude (saved juice, *Equation 4*), elapsed time (incorporating error trials, *Equation 5*), and tuning to specific sequence steps (*Equation 6*, *Figure 5—figure supplement 1*). In all cases, the adaptive progress model explained a higher number of responses, with a significantly higher number of best fits (all $p<1.0 \times 10^{-20}$, z-test for dependent samples), and with significant shifts in $R^2$ distribution towards the adaptive model (all $p<1.0 \times 10^{-7}$, one-sample t-test). We also tested a partial adaptive progress model that included both adaptive and non-adaptive progress regressors in the same regression model (*Equation 8*). This analysis was motivated by the observation that the population activity on spend trials varied to some extent across different sequence lengths (*Figure 4C*), which suggested that adaptation may have been incomplete. The partial adaptation model was also motivated by related approaches in economics (*Koszegi and Rabin, 2006*) that model dependence of individuals' preferences on external reference points using partial adaptation, although with different formalisms. Similar to the full adaptive model, a partial adaptation provided a better fit to the data than the non-adaptive model (*Figure 5—figure supplement 1F*). However, even though some individual responses were better explained by partial adaptation compared to full adaptation (*Figure 5—figure supplement 1G*), and aspects of population activity seemed more consistent with partial adaptation (*Figure 4C*), on average both full adaptation and partial adaptation fit the data equally well (*Figure 5—figure supplement 1F*) and explained similar numbers of responses (full adaptive model: 447/1126 task-related responses, 39.7%; partial adaptive model: 435/1126 responses, 38.6%). Thus, an adaptive model that normalized sequence progress by sequence length provided a suitable characterization of the population of amygdala responses.

We tested whether adaptation to sequence length disappeared in the imperative task, in which sequences were not internally planned but externally instructed, focusing on the subset of activities significant for progress in both tasks. True to this prediction, in the imperative task, the adaptive progress model did not explain a higher number of responses and did not have a higher number of best-fit responses compared with the main alternative models ($p>0.1$). Exceptions were the elapsed time and tuning models ($p<1.0 \times 10^{-4}$), which provided generally poorer fits in the free choice task. Directly contrasting model fits between free and imperative tasks showed that the relative advantage of the adaptive model (quantified by positive $R^2$ difference between models) was significantly larger in the free choice task compared to the imperative task in all cases (*Figure 5D*), except for

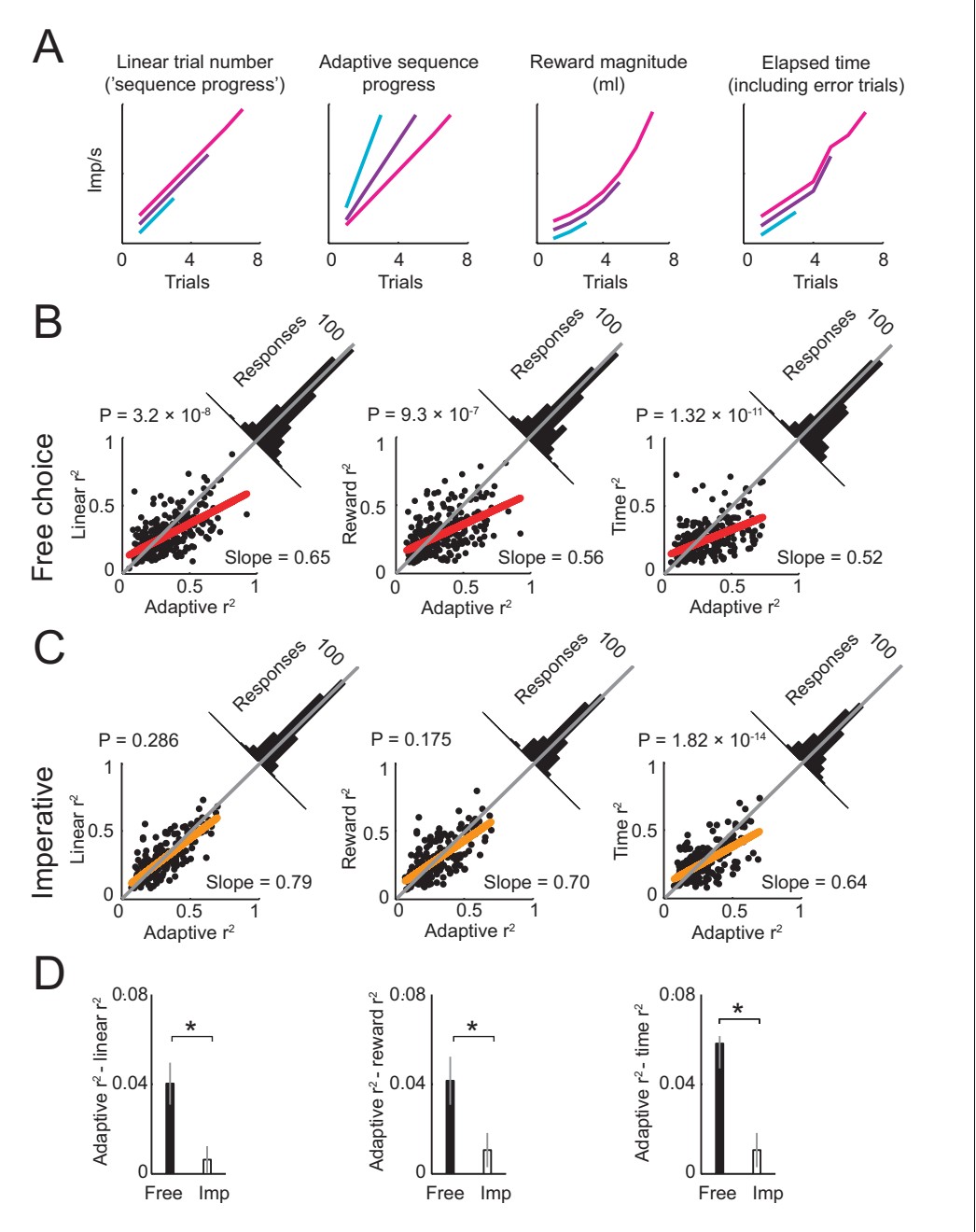

**Figure 5.** Comparing different models of sequence progress. (**A**) Schematic illustration of four models of gradual activity. Left to right: (i) linear non-adaptive sequence progress (cumulative record of correct trials within sequence; *Equation 2*); (ii) adaptive sequence progress (cumulative trial record normalized by the final sequence length; *Equation 3*); (iii) reward magnitude (saved juice amount in current sequence in ml, shown for a typical interest rate; *Equation 4*); (iv) elapsed sequence time (cumulative trial record including error trials; *Equation 5*). The curves are vertically separated for visibility. Values generated by the four models were used as regressors for neuronal activity. (**B**) Model comparison in free choice task. The adaptive progress model provided a significantly better fit compared to all alternatives. Each scatter plot compares the $R^2$ (explained variance) obtained by fitting the adaptive progress model to the $R^2$ obtained by fitting the non-adaptive progress (left), reward magnitude (middle), and elapsed time (right) models. Red lines: linear fits of $R^2$ values for model pairs; deviation from unity line (gray) towards the horizontal axis indicates better fit of adaptive model. Angled histograms show distributions of $R^2$ differences; p-values indicate deviation from normal distribution (Kolmogorov-Smirnov test). Each plot is based on all neuronal responses with significant fits for both compared models. (**C**) Model comparison for the

*Figure 5 continued on next page*

*Figure 5 continued*
imperative task. The adaptation model did not provide better fits compared to non-adaptive and reward magnitude models. Each plot is based on all neuronal responses with significant fits for both compared models. (D) Comparison of differences in model fit ($R^2$) between free choice and imperative tasks. All $R^2$-differences in free choice task but none in the imperative task were significantly greater than zero ($p<0.05$, two-tailed t-test). All differences in free choice task were significantly greater than those in imperative task ($p<0.05$, paired t-test). Each plot is based on all neuronal responses with significant fits for both free choice and imperative task.
The following figure supplements are available for figure 5:

**Figure supplement 1.** Model of neuronal activity based on tuning for specific sequence steps.
**Figure supplement 2.** A single amygdala neuron with gradually increasing activity in both free choice and imperative task.

the tuning model, which was similarly inferior in both tasks (*Figure 5—figure supplement 1*). Thus, we found little evidence for adaptation of progress activity in the imperative task.

We distinguished adaptive progress activities that were specific to the free choice task from other kinds of gradual activities that may reflect simpler reward expectation functions. The neuron in Fig. 5-figure supplement two also showed gradual activity increases throughout each choice sequence; however, this gradual activity also occurred in the imperative task. Moreover, in neither task did the activity adapt to planned sequence length, shown by similar progress regression slopes for short and long sequences. Thus, activity in this neuron was markedly distinct from the adaptive progress activities described above, and more consistent with amygdala reward expectation activities previously reported at shorter, single-trial timescales (*Belova et al., 2007*; *Bermudez and Schultz, 2010*).

Taken together, amygdala neurons with progress activity adapted to internally planned sequence length, consistent with the principles of efficient coding. This neuronal adaptation was largely specific to internally guided choice sequences and distinct from reward expectation.

## Reaction time control analysis

We tested relationships between neuronal activity and reaction times by adding both key touch reactions times and saccade reaction times as regressors to the stepwise regression model (i.e. adding the two reaction time measures to *Equation 7*). Similar to previous studies (*Peck and Salzman, 2014*), activity in some neurons reflected trial-by-trial reaction times (saccadic reaction times: 55/326 neurons, 17%; 65/1126 task-related responses, 6%; key reaction times: 82/326 neurons, 25%; 105/1126 task-related responses, 9%). Such responses may reflect trial-by-trial relationships to the animal's motivation or attentional level (*Peck and Salzman, 2014*). Neuronal responses related to reaction times were more frequent in centromedial than in basolateral amygdala neurons (saccade reaction times: 38 vs. 27; key reaction times: 59 vs. 46; both $p<0.005$; $\chi^2$-test).

Importantly, accounting for reaction times did not explain our main result of neuronal responses related to sequence progress: the stepwise regression with reaction times as covariates identified 176 neuronal responses (15%) in 117 neurons (36%) related to sequence progress. Similar relations to reaction times were found for neurons recorded in the imperative task (saccadic reaction times: 24/155 neurons, 15%; 26/508 task-related responses, 5%; key reaction times: 43/155 neurons, 28%; 62/508 task-related responses, 12%).

Thus, some amygdala neurons showed relationships to trial-by-trial reaction times, potentially reflecting the animal's motivation or attention level. Accounting for these effects did not explain our main finding of amygdala activity related to sequence progress.

## Linear population decoding of sequence progress

To quantify the precision with which the population of amygdala neurons encoded sequence progress, we trained biologically plausible linear classifiers to decode sequence progress steps from aggregated single-trial activities. Cross-validated decoding performance of a nearest-neighbor classifier (*Quian Quiroga et al., 2006*) was significantly above chance in all trial phases (*Figure 6A*; mean accuracy: $78.13 \pm 0.56\%$) with maximum accuracy of 89% in the fixation phase. Similar results

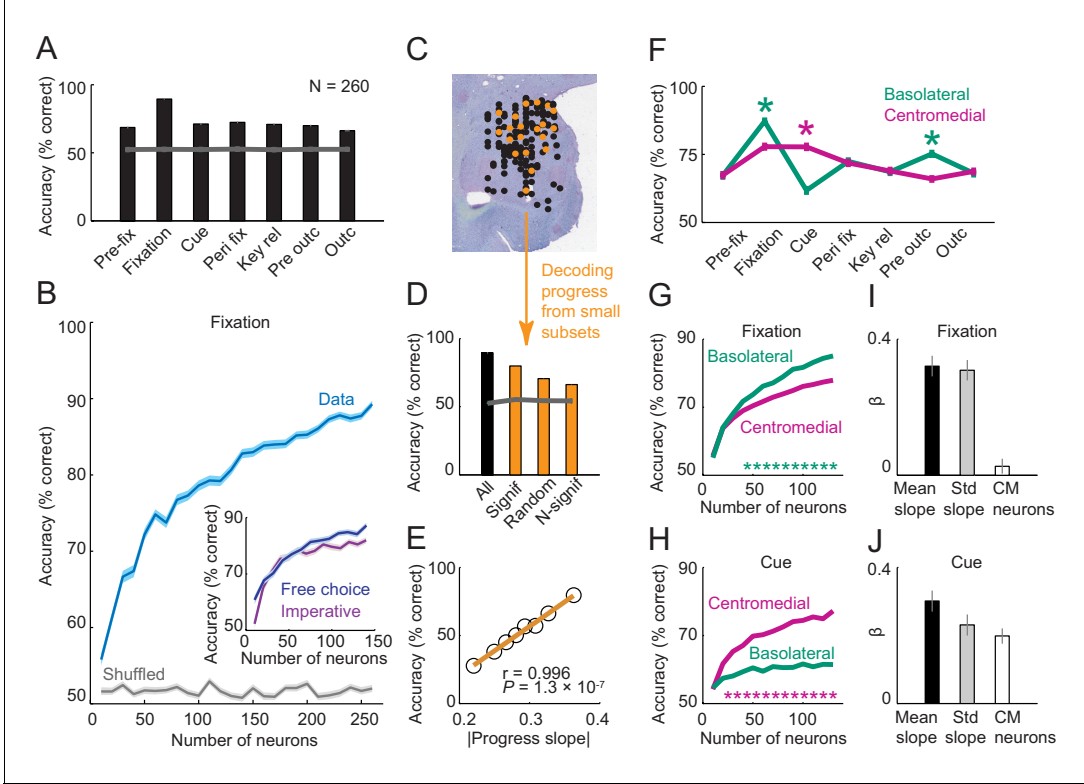

**Figure 6.** Population decoding of sequence progress. (**A**) Decoding accuracy (% correct classification) of a nearest-neighbor classifier. Leave-one-out cross-validated decoding was significantly above decoding from shuffled data (gray line; all p<0.0001; Wilcoxon test). Decoding was based on normalized single-trial activity of all neurons that met criteria of decoding analysis (N = 260) without pre-selection for task-relatedness. (**B**) Performance increased with the number of neurons, shown here for the fixation period. Data for each neuron number show mean (± s.e.m) over 100 iterations of randomly selected neurons. Inset shows decoding for a subset of neurons recorded in both free choice and imperative task. (**C–E**) Decoding sequence progress from small neuronal subsets (N = 20) helped to identify the factors that influenced performance. (**C**) Illustration of subset analysis: Decoding was performed for the fixation period based on sets of 20 randomly selected neurons (illustrated in orange) for defined subpopulations, including (i) only neurons significant for sequence progress ('Signif'; **Equation 2**); (ii) randomly chosen neurons irrespective of significance ('Random'); (iii) only non-significant neurons ('N-signif'). For each subset category, we performed 3000 iterations. (**D**) Performance from defined neuronal subsets (N = 20, orange bars) and from all neurons (N = 260, black bar). In all cases, decoding was significantly above chance. (**E**) Performance depended on individual neurons' progress sensitivities. Linear regression (**Equation 2**) of performance from randomly selected subsets ('Random') on subsets' mean progress slopes. (**F**) Comparison of sequence progress decoding for basolateral (N = 137) and centromedial (N = 123) amygdala neurons (mean ± s.e.m; *p<0.0001, Wilcoxon test). (**G, H**) Increases in decoding performance with number of basolateral and centromedial neurons in fixation (**G**) and cue (**H**) periods (*p<0.0001, Wilcoxon test). (**I, J**) Subset analysis for basolateral and centromedial neurons. Bars show regression coefficients obtained from regressing decoding performance on subsets' mean rectified progress slopes, standard deviation ('Std') of progress slopes, and fraction of centromedial ('CM') neurons in the subset.

The following figure supplement is available for figure 6:

**Figure supplement 1.** Population decoding of sequence progress using support vector machine classifier.

were obtained using a support vector machine classifier (mean accuracy: 83.01% ± 0.92; *Figure 6—figure supplement 1*). We focus our main results on the nearest-neighbor classifier for comparability with our earlier study in which we used nearest-neighbor decoding of choices from amygdala activity (*Grabenhorst et al., 2012*); we confirmed similar or more accurate results using the support vector machine. Decoding performance increased steadily with the number of neurons used for decoding (*Figure 6B*), which suggested that different neurons contributed partly independent information. Interestingly, comparing decoding between free choice and imperative task for neurons recorded in both tasks shows that considerable information about sequence progress was present in amygdala neurons during the imperative task, even though decoding was on average better during the free choice task (*Figure 6B* inset; p=0.033, Wilcoxon test on accuracies for maximum number for neurons

shown in *Figure 6B* inset). This is consistent with the single neuron results which showed reduced numbers of neurons significant for sequence progress in the imperative task compared to the free choice task.

To determine which factors influenced decoding performance, we repeatedly constructed small subsets of 20 randomly selected neurons (*Figure 6C–E*). Decoding based on these small subsets of randomly selected neurons was well above chance (*Figure 6D*, 'Random'). Restricting subsets to those neurons that were individually significant for sequence progress (*Equation 2*) resulted in substantially higher performance, although accuracy was slightly reduced compared to decoding from all neurons (*Figure 6D*, 'Significant' compared to 'All"). This suggested that even individually non-significant neurons contributed to decoding (*Figure 6D*, 'N-signif'). Overall, however, decoding performance depended strongly on individual neurons' sensitivities (slopes) to sequence progress (*Figure 6E*). Thus, the population of amygdala neurons allowed decoding of sequence progress with high accuracy, with individually sensitive neurons contributing most strongly to this population code.

We also compared decoding contributions of basolateral and centromedial amygdala neurons, as these amygdala regions differ in connectivity and function (*Balleine and Killcross, 2006*; *Duvarci and Pare, 2014*; *Mosher et al., 2010*; *Namburi et al., 2015*). For both regions, decoding was well above chance in all task periods (*Figure 6F*). Notably, in the fixation period, decoding performance was significantly higher for basolateral neurons, whereas in the cue period performance was significantly higher for centromedial neurons (*Figure 6F*). Regional differences were also evident when adding different numbers of neurons to the classifier. In the fixation period, adding more neurons increased performance for both basolateral and centromedial neurons, with overall higher accuracy for basolateral neurons (*Figure 6G*). By contrast, in the cue period, accuracy increased substantially as more centromedial neurons were added, with significantly lower accuracy increases for basolateral neurons (*Figure 6H*). Subset analyses confirmed the relative importance of centromedial neurons for cue-phase decoding. In the fixation period, the relative number of centromedial neurons in a decoding subset had no impact on performance (*Figure 6I*). By contrast, in the cue phase, adding more centromedial neurons (relative to basolateral neurons) significantly increased decoding accuracy, even irrespective of their progress sensitivities, which were covariates in the analysis (*Figure 6J*). Thus, although neurons from both basolateral and centromedial amygdala allowed accurate progress decoding, there were significant regional differences between fixation and cue periods, potentially related to distinct behavioral requirements of these task periods (e.g. internal valuation during fixation compared to response to external events during the cue period).

## Discussion

We identified gradual, stepwise responses in amygdala neurons as monkeys executed economic choice sequences according to internal plans. These responses occurred in the absence of external progress cues and without opportunities to preplan specific actions. They were often specific for internally guided choice sequences, despite similar behavioral reward expectation in free choice and imperative control task. The slope of the gradual responses reflected prospectively the forthcoming choice sequence lengths, readily from one sequence to the next, suggesting adaptation to the animal's internal behavioral plan. Accordingly, gradual amygdala activity was best described in terms of adaptive sequence progress (normalized by planned sequence length), rather than by reward expectation or timing models. Sequence progress activity fluctuated with performance as it was diminished on error trials and subsequently reinstated, suggesting the behavioral relevance of progress signals. Linear population decoding suggested a progress code that can be read out with high accuracy by downstream neurons. These gradual responses contrast markedly with other amygdala activities observed in the same reward-saving task: rather than directly predicting single-trial choices or choice sequences (*Grabenhorst et al., 2012*; *Hernadi et al., 2015*), the present activities changed dynamically over consecutive trials as a function of sequence progress. Such continual evaluation of progress in amygdala neurons seems crucial for aligning individual choices with behavioral plans and for successfully navigating toward reward goals (*Benhabib and Bisin, 2005*; *Berkman and Lieberman, 2009*; *Johnson and Busemeyer, 2001*). These data implicate gradual amygdala responses in the evaluation of progress during internally planned choice sequences and support a role for the primate amygdala in advanced economic behaviors.

Amygdala neurons respond to reward-predictive stimuli (*Nishijo et al., 1988*; *Paton et al., 2006*; *Rolls, 2000*; *Sugase-Miyamoto and Richmond, 2005*), encode related reward expectation (*Belova et al., 2007*; *Bermudez and Schultz, 2010*), and participate in regulating affective state and arousal in response to external events (*Davis and Whalen, 2001*; *Janak and Tye, 2015*; *LeDoux, 2000*). The majority of the presently described progress-related activities are probably not directly derived from these basic, reactive reward functions. First, in many neurons, the responses occurred specifically during freely determined choice sequences but not in closely matched imperative sequences. Notably, reaction time patterns were similar between both tasks, suggesting matched behavioral reward expectation. Second, gradual activity was better described by an adaptive progress model than by explicit reward expectation or timing. This neuronal adaptation was also specific to the free choice task, despite behavioral reaction time adaptation in both tasks. A minority of neurons showed non-adaptive gradual activity in both free choice and imperative tasks, potentially reflecting reward expectation or affective state regulation. In a previous study, primate amygdala neurons during visually guided reward schedules responded to sequence onset, aspects of the cued schedule, and reward delivery, without tracking schedule progress (*Sugase-Miyamoto and Richmond, 2005*). We suggest that our internally guided choice task, which allowed the animals to plan behavior over several steps, potentially engaged amygdala neurons to a greater extent than purely visually instructed tasks. The lower percentage of progress activities we observed in the imperative task is consistent with this interpretation. Thus, although activity in some neurons matched known amygdala reward expectation signals, the majority of amygdala progress activities reported here were unlikely due to basic reward expectation or affective reactions to external stimuli.

The amygdala contains heterogeneous cell types with varying projection targets (*Janak and Tye, 2015*; *LeDoux, 2000*; *Namburi et al., 2015*). Accordingly, the functional significance of the present gradual activities likely depends on a neuron's precise outputs. Amygdala progress-related activity during choice sequences could provide inputs for decision computations (*Deco et al., 2013*; *Wang, 2002*) in cortical areas to which the amygdala projects. This could apply especially to basolateral neurons with projections to anterior cingulate cortex, where gradual activity changes occur in multistep reward schedules (*Shidara and Richmond, 2002*) and foraging tasks (*Hayden et al., 2011*), and to orbitofrontal cortex, where neurons signal economic values during single-trial choices (*Kennerley et al., 2009*; *Padoa-Schioppa, 2009*; *Padoa-Schioppa and Assad, 2006*). Consistent with this suggestion, reward value coding in orbitofrontal cortex depends partly on the integrity of the amygdala (*Rudebeck and Murray, 2008*). In amygdala neurons with striatal projections, gradual activity could potentially exert a more direct influence on action, as amygdala-striatal connections are implicated in reward-seeking behavior (*Namburi et al., 2015*; *Stuber et al., 2011*; *Tye et al., 2008*). This possibility seems consistent with the behavioral consequences of amygdala lesions (*Baxter and Murray, 2002*; *Bechara et al., 1999*; *Brand et al., 2007*). Although amygdala inactivation does not cause deficits in object choice once reward values have been updated (*Wellman et al., 2005*), the amygdala is thought to make essential contributions to reward-guided behavior in situations that required dynamic valuations (*Murray and Rudebeck, 2013*). Furthermore, gradual activity in amygdala neurons with outputs to attentional and autonomic effector systems (*Peck et al., 2013*; *Price, 2003*) could serve to progressively focus cognitive and motivational resources during internally directed choice sequences. These interpretations of the behavioral relevance of amygdala progress signals are supported in the present study by relationships between progress signals and performance on error trials, and by relationships between some amygdala neurons and the animals' reaction times. Finally, we suggest that gradual activity in amygdala neurons with predominantly local connections could participate in putative decision computations within local amygdala circuits. From this perspective, the progress-related amygdala activity in the current task could be interpreted as reflecting the animal's gradually increasing intention to make a spend choice. This suggestion is supported by the existence of explicit spend choice signals in amygdala neurons (*Grabenhorst et al., 2012*) (which reflect the output of a decision computation), decision deficits in humans with amygdala lesions (*Bechara et al., 1999*; *Brand et al., 2007*), decision-related activations in human imaging studies (*De Martino et al., 2006*; *Grabenhorst et al., 2013*), and with features of amygdala inhibitory microcircuits (*Duvarci and Pare, 2014*; *Janak and Tye, 2015*) that could support neural choice mechanisms by winner-take-all competition through mutual inhibition (*Deco et al., 2013*; *Wang, 2002*). Although some amygdala neurons showed lasting, tonic activity

modulation by sequence progress, most progress responses were phasic and time-locked to individual task events, suggesting they may have played specific roles during task performance, rather than reflect lasting changes in the animal's state. Overall, gradual activity patterns in amygdala neurons could reflect a number of underlying processes related to behavioral control, decision-making, and the regulation of motivation, attention and affective state. This interpretation is supported by our finding that some individual neurons and population activity also exhibited gradual patterns in the imperative task, which likely reflected reward expectation and related processes. However, our main result that progress activity in a group of amygdala neurons was adapted to internal plans and occurred specifically during free choices seems more closely linked to internally guided decision processes. Testing these possibilities will likely require a circuit-based approach, e.g. using optogenetics tools, to examine the presently characterized gradual responses in amygdala subpopulations with defined projection targets (*Janak and Tye, 2015*; *Namburi et al., 2015*).

In various brain systems, the gradual evolution of mental processes towards a decision is accompanied by gradually increasing, 'ramping' activity. This ramping activity is thought to reflect sensory evidence accumulation or motor response preparation through integration-to-bound or evidence accumulation mechanisms (*Hanes and Schall, 1996*; *Maimon and Assad, 2006*; *Mazurek et al., 2003*; *Murakami et al., 2014*; *Okano and Tanji, 1987*; *Roitman and Shadlen, 2002*; *Romo and Schultz, 1987*; *Schultz and Romo, 1988*; *Shadlen and Newsome, 2001*; *Stuphorn et al., 2010*). Similar ramping activities also occur in reward structures including monkey striatum and amygdala, where neurons show ramping activity to reward-predicting cues on single trials (*Bermudez and Schultz, 2010*; *Schultz, 2015*). Our results differ from these previous studies in important respects. In contrast to perceptual decisions or motor preparation, our task did not provide external stimuli to elicit evidence accumulation nor did it allow preplanning of specific motor responses. Furthermore, the presently reported gradual amygdala activity is morphologically different from ramping activity: it was not confined to the timescale of individual trials but reoccurred as a phasic response during choice sequences lasting up to two minutes. Finally, although resembling gradual activity observed in anterior cingulate cortex during foraging (*Hayden et al., 2011*), the present gradual amygdala responses were mostly specific free choice situations (an aspect not tested in the previous study) and adapted prospectively to the monkey's internal behavioral plan, rather than monitoring value changes of a foraging patch in the external environment.

The observed prospective adaptation to internally planned sequence length in the present amygdala neurons seems important to optimize neuronal discrimination of different sequence steps. Accurate signaling of progress is critical for guiding step-by-step choices according to internal plans and for regulating motivation during the pursuit of reward goals (*Berkman and Lieberman, 2009*). Adaptation is a fundamental process that enables accurate coding of variables by neurons with a limited activity range (*Fairhall et al., 2001*; *Laughlin, 1981*; *Schultz, 2015*). Across neurons, our data were consistent with both full adaptation and partial adaptation to planned sequence length. Conceivably, incomplete adaptation in some neurons may enhance flexibility of sequence progress encoding across a neuronal population by retaining some information about the absolute, non-adapted sequence length. Our model of partial adaptation was motivated by related approaches in economics (*Koszegi and Rabin, 2006*) that model dependence of individuals' preferences on external reference points using partial adaptation, although with different formalisms. Our population decoding results, obtained using linear, biologically plausible classifiers, confirm the accuracy of progress coding across the population of amygdala neurons. Previously reported amygdala neurons signaling length and value of planned sequences (*Hernadi et al., 2015*) seem well suited to guide this prospective adaptation, although the underlying neurophysiological mechanism remains unclear. As our neurophysiological experiments were necessarily restricted to choice sequences lasting up to several minutes, our data do not provide evidence concerning planned behaviors over even longer timescales, which likely involve additional mechanisms.

Achieving rewards in the real world typically requires sequences of choices, often according to an internal plan (*Benhabib and Bisin, 2005*; *Berkman and Lieberman, 2009*; *Johnson and Busemeyer, 2001*). Our data indicate that amygdala neurons participate in the evaluation of progress during internally planned choice sequences. Given the amygdala's diverse anatomical outputs, such progress signals could potentially support a range of behavioral functions, including multistep decision-making and the focusing of motivation, affective state and cognitive processes on self-defined goals. The present results may also shed new light on the behavioral and mental deficits of humans with

psychiatric conditions involving amygdala dysfunction (*Koob and Volkow, 2010*; *Price and Drevets, 2010*). Future studies could test whether impaired progress evaluation by the amygdala might contribute to deficits in reward pursuit and goal-directed behaviors in these conditions.

## Materials and methods

### Neurophysiological recordings

All animal procedures conformed to the US National Institutes of Health Guidelines and were approved by the Home Office of the United Kingdom (Home Office Project Licenses PPL 80/2416, PPL 70/8295, PPL 80/1958, PPL 80/1513). Experimental procedures for neurophysiological recordings from awake behaving macaque monkeys have previously been described (*Bermudez and Schultz, 2010*; *Grabenhorst et al., 2012*). Two adult male rhesus monkeys (Macaca mulatta), weighing 9.2 and 12.0 kg, participated in the present experiments. A head holder and recording chamber were fixed to the skull under general anesthesia and aseptic conditions. The number of animals used is typical for primate neurophysiology experiments. We used bone marks on coronal and sagittal radiographs to localize the anatomical position of the amygdala in reference to the stereotaxically implanted chamber (*Aggleton and Passingham, 1981*). We recorded single-neuron activity extracellularly, using standard electrophysiological techniques, from dorsal, lateral, and basal amygdala.

In exploratory tests, we sampled activity from about 700 amygdala neurons and recorded and saved the activity of neurons that appeared to respond to any task event during online inspection of several trials. Thus, we aimed to identify task-related neurons but we did not preselect based on more specific response characteristics. This procedure resulted in a database of 326 neurons which we analyzed statistically. The number of neurons is similar to those reported in previous studies on primate amygdala. Statements about the number of neurons showing specific effects are made with reference to these task-related neurons.

Following completion of data collection, we made electrolytic lesions (15–20 μA, 20–60 s) to mark recording sites for histological reconstruction (*Figure 3F,G*). The animals received an overdose of pentobarbital sodium (90 mg/kg iv) and were perfused with 4% paraformaldehyde in 0.1 M phosphate buffer through the left ventricle of the heart. We reconstructed recording positions from 50-μm-thick, stereotaxically oriented coronal brain sections stained with cresyl violet.

### Sequential choice task

Monkeys chose on each trial to save (accumulate) the liquid reward that was available on that trial, which increased its magnitude by a variable 'interest rate', or spend (consume) the saved amount. Increase in reward magnitude over sequential save choices were governed by a geometric series (*Equation 1*)

$$x_n = b \sum_{i=0}^{n-1} q^i \tag{1}$$

with $x_n$ as reward magnitude on trial $n$, $b$ as base rate of reward magnitude, and $q$ as interest rate. Choice sequences were self-determined in that monkeys were free to produce save-spend sequences of different lengths (following one required save choice per sequence). For high interest rates, we delivered a fixed amount of 8 ml after seven consecutive save trials, as this was the maximum amount the animals could consume on one trial. The animals were still free to produce longer saving sequences (no imposed upper limit on sequence length). After training, the animals only generated sequences that resulted in reward amounts that they could comfortably drink.

A computer-controlled solenoid valve delivered juice reward from a spout in front of the animal's mouth. For monkey A the base rate of reward magnitude, b from *Equation 1*, was set to 0.11 ml for all sessions, for monkey B the base rate was set to 0.11 ml for half of the sessions and 0.13 ml for the other half of the sessions. The animal's tongue interrupted an infrared light beam below the adequately positioned spout. An optosensor monitored licking behavior with 0.5 ms resolution (STM Sensor Technology).

The animals initiated trials by placing their hand on an immobile, touch-sensitive key, followed by presentation of an ocular fixation spot on a computer monitor (1.3° visual angle). The animals were

required to fixate within 2–4° for 1500 ms plus mean of 500 ms (truncated exponential distribution). We monitored eye position using an infrared eye tracking system at 125 Hz (ETL200; ISCAN). Two save and spend visual stimuli of 7.0° then appeared on the left and right side of the monitor (pseudorandomized). In blocks of typically 40–100 consecutive trials, we used different pre-trained stimuli as save cues to indicate different interest rates. (Each neuron was typically tested with one to two different interest rates. The duration required for testing neurons with statistically sufficient numbers of trials in both free choice and imperative tasks usually precluded using more than two interest rates) Animals could indicate their choice with a saccade towards the save or spend cue as soon as the cues appeared. The chosen stimulus was replaced by a peripheral fixation spot of 7.0° of visual angle. Following a delay of 1500 ms, the peripheral fixation spot changed color and signaled the animal to release the touch key which resulted in delivery of the reinforcer (auditory or visual cue on save trials vs. a liquid reward on spend trials). Crucially, there were no external cues that signaled sequence progress to the animals. Thus, the animals were required to track progress internally. Failures of key touch or fixation resulted in trial cancellation; more than three sequential errors led to a pause in behavioral testing. Accumulated saved rewards were retained across error trials. The animals were overtrained by the time of neuronal recording and showed consistent, meaningful saving behavior for different interest rates without further signs of learning.

To provide an example of how rewards were calculated, consider a series of two successive save choices by the monkey with a base rate of reward b = 0.11 and interest rate q = 1.5. On the second trial of the choice sequence, after the first save choice, reward R = 0.11 × (1 + 1.5) = 0.275 ml. On the third trial, after two successive save choices, reward R = 0.11 × (1 + 1.5 + $1.5^2$) = 0.523 ml.

## Imperative control task

In the imperative control task, the animals performed behavioral sequences of matched lengths to the free choice task. However, the behavior was not self-controlled but was externally instructed by a small visual cue presented next to either the save or the spend cue. The instruction cue indicated the correct choice on each trial. Trials were otherwise identical to a free choice trial. We matched the ratio of save to spend trials between imperative and free choice task for each monkey and interest rate to allow the animals to anticipate reward amounts which we confirmed by behavioral reaction times (*Figure 1—figure supplement 1*). The imperative task was performed for a subset of recorded neurons in a separate trial block. For these neurons, the tasks were on average counterbalanced so that either the free choice or imperative task was recorded first. Although we matched sequence lengths between both tasks for specific animals and interest rates, the counterbalanced order meant that for some sessions sequence lengths were not perfectly matched (e.g. if the imperative task was performed first and the animal subsequently produced a larger or smaller range of sequences during free choice). However, this is unlikely to be a major factor in our results as mean sequence lengths were highly correlated between pairs of free choice and imperative sessions (R = 0.527, p=2.1 × $10^{-12}$) and sequence length ranges did not differ significantly on average (p=0.949, t-test).

## Data analysis

### Linear regression analysis of reaction times and licking durations

As a measure of the animals' trial-by-trial reward expectation, we analyzed the latencies with which the monkeys released the touch key at the end of the trial to initiate reinforcer delivery. Reaction times were z-normalized separately for each animal within each experimental session by subtracting the session mean and dividing by the session standard deviation.

### Analysis of neuronal data

We counted neuronal impulses in each neuron on correct trials in fixed time windows relative to different task events: 1000 ms before fixation spot (Pre-fixation), 1775 ms after fixation spot but before cues (Fixation, starting 25 ms after fixation spot onset), 300 ms after cues (Cue, starting 20 ms after cue onset), 1500 ms post-choice delay (Delay, starting 25 ms after the animal had indicated its choice), and 500 ms during the reward/outcome period (Outcome, starting 50 ms after reinforcer onset).

We first identified task-related responses in individual neurons and then used linear regression analysis to identify neuronal responses related to sequence progress. We identified task-related responses by comparing activity in the Fixation, Cue, Delay and Outcome periods to a control period (Pre-fixation) using the Wilcoxon test ($p<0.0083$, Bonferroni-corrected for multiple comparisons). A neuron was included as task-related if its activity in at least one task period was significantly different to that in the control period. Because the Pre-fixation period served as control period we did not select for task-relatedness in this period and included all neurons with observed impulses in the analysis. We chose the pre-fixation period as a control period because it was the earliest period at the start of a trial in which no sensory stimuli were presented. Our analysis strategy was to first examine gradual amygdala responses with a single linear regression model to determine how many responses could be described in terms of a linearly increasing (or decreasing) activity pattern. We used single linear regression in fixed time windows around task events as this provided the most direct way to detect and interpret such gradual activity patterns, and to allow direct comparison with results from previous studies using similar sequential choice tasks (*Hayden et al., 2011*). We used different single linear regressions to distinguish whether gradual activities were best described in terms of linear increase, or various non-linear increases related to task variables of interest. We also used a multiple stepwise regression model (*Equation 7*) as an alternative analysis to determine the number of gradual responses when other task-relevant variables were accounted for. We also used additional sliding window regressions in combination with a bootstrap method without pre-selecting task-related responses to confirm that our results did not depend on the pre-selection of task-related responses or definition of fixed analysis windows. Finally, population decoding examined independence of our findings from pre-selection of task-related responses and served to assess information about gradual activity contained in the neuronal population. The statistical significance of regression coefficients was determined with a t-test using $p<0.05$ as criterion. All tests performed were two-sided. Each neuronal response was tested with the following regression models:

$$y = \beta_0 + \beta_1 \, Sequence\,progress + \epsilon, \tag{2}$$

$$y = \beta_0 + \beta_1 \, Sequence\,progress/Sequence\,length + \epsilon, \tag{3}$$

$$y = \beta_0 + \beta_1 \, Reward\,magnitude + \epsilon, \tag{4}$$

$$y = \beta_0 + \beta_1 \, Elapsed\,time + \epsilon, \tag{5}$$

$$y = \beta_0 + \beta_1 \, Step1 + \beta_2 \, Step2 + \ldots + \beta_n \, StepN + \epsilon, \tag{6}$$

$$y = \beta_0 + \beta_1 \, Choice + \beta_2 \, SVspend + \beta_3 \, SVsave + \beta_4 \, SeqSV + \beta_5 \, SeqLength + \beta_6 \, Cue\,position + \\ \beta_7 \, Left/right + \beta_8 \, Sequence\,progress + \epsilon, \tag{7}$$

$$y = \beta_0 + \beta_1 \, Sequence\,progress + \beta_2 \, Sequence\,progress/Sequence\,length + \epsilon, \tag{8}$$

with $y$ as trial-by-trial neuronal impulse rate, *Sequence progress* as sequence progress defined by the current cumulative number of trials in a sequence, *Sequence progress/Sequence length* as sequence progress normalized by forthcoming sequence length, *Reward magnitude* as reward magnitude in ml calculated according to (*Equation 1*), *Elapsed time* as elapsed time calculated according to sequence progress including error trials, *Step1 to StepN* as specific steps in a saving sequence, *Choice* as current-trial save-spend choice, *SeqSV* as sequence value, *SeqLength* as sequence length, *Cue position* as current-trial left-right position of the save cue, and *Left/right* as an indicator function denoting whether the monkey made a saccade to the left or to the right. Value regressors for spend value, save value and sequence value were derived from the animals' observed choice probabilities for different sequence lengths as described previously (*Hernadi et al., 2015*). The model in (*Equation 7*) was tested both using a stepwise approach and standard (simultaneous) estimation of coefficients as described in the main text and *Figure 3—figure supplement 1*.

## Definition of subjective values

In *Equation 7* we included subjective values related to save and spend choices and whole saving sequences as control regressors for sequence progress. These subjective values were previously shown to provide a good description of the animals' saving behavior (*Grabenhorst et al., 2012*; *Hernadi et al., 2015*). Subjective values were derived for each interest rate from the relative spending frequency at each step in a saving sequence, multiplied by the associated objective reward magnitude (to account for differences in reward magnitude between interest rates). This measure constituted the subjective value of spending on each trial ('spend value'). Thus, the subjective value for spending, SVspend, at a given point i in a saving sequence was defined as (*Equation 9*)

$$SVspend_i = P_i M_i, \tag{9}$$

where $P_i$ is the probability with which the monkey produced a saving sequence of length $i$, and $M_i$ is the objective reward magnitude in ml of juice that would result from spending at point $i$ of the sequence length given the current interest rate. The spend value actually realized in a saving sequence constituted the value of the current sequence ('*SeqSV*'). The 'save value' for each trial was defined as the average spend value that could be obtained in all future trials of a sequence. Thus, the subjective value *SVsave* for saving at a given point $n$ in a save sequence was defined as (*Equation 10*)

$$SVsave_n = \frac{1}{m-n} \sum_{i=n+1}^{m} SVspend_i, \tag{10}$$

with $m$ defining the upper limit of the save sequence (given by the maximal observed sequence length for the monkey).

## Sliding window regression analysis of neuronal data

We used a sliding window regression analysis with a 200 ms window that we moved in steps of 25 ms across each trial (*Figure 3E*). To determine whether neuronal activity was significantly related to sequence progress we used a bootstrap approach based on shuffled data as follows. For each neuron, we performed the sliding window regression 1000 times on trial-shuffled data and determined a false positive rate by counting the number of consecutive windows in which a regression was significant with $p<0.05$. We found that less than five per cent of neurons with trial-shuffled data showed more than eight consecutive significant analysis windows. Therefore, we counted a sliding window analysis as significant if a neuron showed a significant ($p<0.05$) effect for more than eight consecutive windows.

## Normalization of population activity

We subtracted from the measured impulse rate in a given task period the mean impulse rate of the control period and divided by the standard deviation of the control period (z-score normalization). Next, we distinguished neurons that showed a positive relationship to sequence progress and those with a negative relationship, based on the sign of the regression coefficient, and sign-corrected responses with a negative relationship. Normalized data were used for *Figure 4C–I*, *Figure 6*, and *Figure 6—figure supplement 1*.

## Normalization of regression coefficients

Standardized regression coefficients were defined as $x_i(s_i/s_y)$, $x_i$ being the raw slope coefficient for regressor i, and $s_i$ and $s_y$ the standard deviations of independent variable i and the dependent variable, respectively. Standardized regression coefficients were used for *Figure 3C*, *Figure 5B C*, *Figure 6E,I,J*.

## Population decoding

We used nearest-neighbor (NN) and support vector machine (SVM) classifiers (*Figure 6*, *Figure 6—figure supplement 1*) to quantify the information about sequence progress contained in neuronal population activity in defined task periods, following decoding analysis approaches from previous neurophysiological studies (*Hung et al., 2005*; *Quian Quiroga et al., 2006*). The SVM classifier was

trained on a set of training data to find a linear hyperplane that provided the best separation between patterns of neuronal population activity defined by a grouping variable based on sequence progress. Decoding was typically not improved by non-linear (e.g. quadratic) kernels. The NN classifier was similarly trained on a set of test data and decoding was performed by assigning each trial to the group of its nearest neighbor in a space defined by the distribution of impulse rates for the different levels of the grouping variables using the Euclidean distance (Quian Quiroga et al., 2006). Both SVM and NN classification are biologically plausible in that a downstream neuron could perform similar classification by comparing the input on a given trial with a stored vector of synaptic weights. Both classifiers performed qualitatively similar, although SVM decoding was typically more accurate.

We aggregated z-normalized trial-by-trial impulse rates of independently recorded amygdala neurons from specific task periods into pseudo-populations. We used all recorded neurons that met inclusion criteria for a minimum trial number, without pre-selecting for value coding, except where explicitly stated. We only included neurons in the decoding analyses that had a minimum number of five trials per group for which decoding was performed, and we confirmed that results were very similar when changing this minimum number to 10 trials. For each decoding analysis, we created three n by m matrices with n columns determined by the number of neurons and m rows determined by the number of trials. We defined three matrices, one for each group for which decoding was performed. These included separate groups for low, medium, and high sequence progress, determined for each neuron by calculating sequence progress terciles. (We obtained very similar results by repeating the decoding analyses based on progress quartiles) Thus, each cell in a matrix contained the impulse rate from a single neuron on a single trial measured for a given group. Because neurons were not simultaneously recorded, we randomly matched up trials from different neurons for the same group and then repeated the decoding analysis with different random trial matching (within-group trial matching) 150 times. We found these repetition numbers to produce very stable classification results. (We note that this approach likely provides a lower bound for decoding performance as it ignores potential contributions from cross-correlations between neurons; investigation of cross-correlations would require data from simultaneously recorded neurons) We used a leave-one-out cross-validation procedure whereby a classifier was trained to learn the mapping from impulse rates to groups on all trials except one; the remaining trial was then used for testing the classifier and the procedure repeated until all trials had been tested. The results were very similar by using 80% of trials as training data and 20% as test data.

The SVM decoding was implemented in Matlab (Version R2013b, Mathworks, Natick, MA) using the 'svmtrain' and 'svmclassify' functions with a linear kernel and the default sequential minimal optimization method for finding the separating hyperplane. Decoding could typically not be improved by using radial basis function or quadratic kernels. The NN decoding was performed in Matlab using custom-written code. We quantified decoding accuracy as the percentage of correctly classified trials, averaged over all decoding analyses for different random within-group trial matchings. To investigate how decoding accuracy depended on population size, we randomly selected a given number of neurons at each step and then determined the percentage correct. For each step (i.e. each possible population size) this procedure was repeated 100 times. We also performed decoding for randomly shuffled data (shuffled group assignment without replacement) with 1500–5000 iterations to test whether decoding on real data differed significantly from chance. Statistical significance was determined by comparing vectors of percentage correct decoding accuracy between real data and randomly shuffled data using the rank sum test (Quian Quiroga et al., 2006). For all analyses, decoding was performed on neuronal responses taken from the same task period.

## Acknowledgements

We thank Ken-Ichiro Tsutsui for help with task design; Anthony Dickinson, John Assad, Raymundo Báez-Mendoza, Armin Lak, William Stauffer, and Martin O'Neill for helpful discussions; Mercedes Arroyo for histology. We also thank the Wellcome Trust, the European Research Council (ERC), and the National Institutes of Health Caltech Conte Center for financial support.

## Additional information

### Competing interests

WS: Reviewing editor, *eLife*. The other authors declare that no competing interests exist.

### Funding

| Funder | Grant reference number | Author |
| --- | --- | --- |
| Wellcome Trust | | Wolfram Schultz |
| European Research Council | | Wolfram Schultz |
| National Institutes of Health | Caltech Conte Center | Wolfram Schultz |

The funders had no role in study design, data collection and interpretation, or the decision to submit the work for publication.

### Author contributions

FG, WS, Conception and design, Analysis and interpretation of data, Drafting or revising the article; IH, Conception and design, Acquisition of data

### Author ORCIDs

Fabian Grabenhorst, http://orcid.org/0000-0002-6455-0648
Istvan Hernadi, http://orcid.org/0000-0001-7882-4817
Wolfram Schultz, http://orcid.org/0000-0002-8530-4518

### Ethics

Animal experimentation: All animal procedures conformed to US National Institutes of Health Guidelines and were approved by the Home Office of the United Kingdom (Home Office Project Licenses PPL 80/2416, PPL 70/8295, PPL 80 / 1958, PPL 80 / 1513).

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
