## [Decision Letter]

Thank you for submitting your article "Primate amygdala neurons evaluate the progress of self-defined economic choice sequences" for consideration by *eLife*. Your article has been favorably evaluated by David Van Essen as the Senior Editor and three reviewers, one of whom, Joshua Gold, is a member of our Board of Reviewing Editors.

The reviewers have discussed the reviews with one another and the Reviewing Editor has drafted this decision to help you prepare a revised submission.

Summary:

The authors report on single-unit activity recorded from amygdala neurons in monkeys performing a sophisticated and novel decision-making task that requires them to plan a sequence of save-and-spend decisions at different interest rates. They found that nearly half of the 326 task-responsive neurons recorded showed gradually increasing activity as the sequence progressed in length, peaking when the animals chose the spend option and received their accumulated reward. Moreover, rate increases in activity adapted to the choice sequence length, consistent with principles of adaptive coding. This pattern of activity was not observed when spend or save decisions were externally instructed. Finally, the authors show that sequence progress can be linearly decoded, suggesting it is possible for a progress code to be read out by downstream areas/neurons.

The reviewers all agreed that the manuscript is technically sound and should be of wide interest. Moreover, it represents an important step in demonstrating the amygdala contributes to more sophisticated computations beyond processing of threat or reward.

Essential revisions:

1) It is not entirely clear which exact regression model (Eqs. 2-7) was used for each analysis presented in the text and figures. This should at least be clarified, and the use of multiple models better justified. Alternatively, it is strongly suggested that the most complete model (Eq. 7) should be used throughout, because it accounts for a number of potentially confounding variables. That model should also be explained better, including descriptions of how *CVspend* and *SVsave* were computed.

2) Several questions were raised about the RT data and their implications. Are the reported RTs on the choice versus imperative task statistically indistinguishable? Such a finding would certainly bolster the argument that reward expectations are similar under the two conditions and thus cannot account for differences in neural activity. If they were different, however (which would not be terribly surprising, given the reported R^2^ values), it should be noted and the implications discussed. Additionally, the reported RTs were computed based on the latency of key press release after a delay period that followed the animal indicating its choice with a saccade. These clearly indicate the behavioral relevance of sequence progress. Is a similar pattern observed for the saccadic RTs? Do they differ on the free choice and imperative tasks? Do they relate to neural activity on the two tasks? Is this relationship similar or different for basolateral and centromedial neurons?

3) Given the strong relationship between neural activity and progress on the choice task, it raises an interesting question about the activity on error trials. Do cells that show pre-fixation or fixation period progress-related activity increases/decreases exhibit aberrant activity on error trials? Or do other aspects of neural activity predict error trials?

4) What was the accuracy of decoding in the imperative control task? A similar figure as Figure 6 for comparison, along with a discussion of the implications of this analysis, would be useful.

5) The observation of "adaptive encoding" is quite interesting. However, according to Figure 4, the adaptation seems only partial in that the normalized maximum firing rate is different for different sequence progress. Have the authors tested a partial adaption model, or adaptive models for reward magnitude and elapsed time (Figure 5)?

6) From inspection of Figure 3, it appears that the majority of the sequence progress activity corresponded to a single, transient modulation that occurred at random times during a trial. The implications of this finding should be at least discussed.

---

## [Author Response]

*1) It is not entirely clear which exact regression model (Eqs. 2-7) was used for each analysis presented in the text and figures. This should at least be clarified, and the use of multiple models better justified. Alternatively, it is strongly suggested that the most complete model (Eq. 7) should be used throughout, because it accounts for a number of potentially confounding variables. That model should also be explained better, including descriptions of how CVspend and SVsave were computed.*

We have now clarified throughout the Results and figure legends which specific regression model was used for each analysis by referring to the specific equation. We have also justified our use of multiple models by explaining our analysis strategy in the Methods, section ‘Analysis of neuronal data’, second paragraph. Briefly, we initially focus on the single linear regression model as it provided the most direct and intuitive way to detect and interpret gradual activity patterns and allowed direct comparison with previous studies using similar sequential tasks (Hayden et al., 2011). However, to be very clear about the numbers of neurons with specific effects, at the start of the Results we now directly compare the numbers of significant responses obtained with the single linear model with our most complete multiple regression model (section ‘Encoding of sequence progress by individual amygdala neurons’). We also provide a more comprehensive explanation of the multiple regression model as requested, including a new section on subjective value definition (Methods, final paragraph of section ‘Analysis of neuronal data’ and subsequent section ‘Definition of subjective values.’).

*2) Several questions were raised about the RT data and their implications. Are the reported RTs on the choice versus imperative task statistically indistinguishable? Such a finding would certainly bolster the argument that reward expectations are similar under the two conditions and thus cannot account for differences in neural activity. If they were different, however (which would not be terribly surprising, given the reported R^2^ values), it should be noted and the implications discussed. Additionally, the reported RTs were computed based on the latency of key press release after a delay period that followed the animal indicating its choice with a saccade. These clearly indicate the behavioral relevance of sequence progress. Is a similar pattern observed for the saccadic RTs? Do they differ on the free choice and imperative tasks? Do they relate to neural activity on the two tasks? Is this relationship similar or different for basolateral and centromedial neurons?*

We performed additional behavioral and neuronal analyses and included new results and figures to address these questions (Results, sections ‘Sequential choice task and economic behavior’, fourth paragraph and ‘Reaction time control analysis’; Figure 1—figure supplement 1, panels C and D). Briefly, statistical examination with multiple regression (Figure 1—figure supplement 1) showed that a task indicator variable (free choice vs. imperative) did not explain additional variance in key touch reaction times compared to other task-relevant variables (neither did interaction effects between the task indicator and sequence progress). This result suggests that the influence of our main variable sequence progress on reaction times did not depend on task type, suggesting similar behavioral relevance for sequence progress across tasks. With respect to neuronal data and reaction times, some neuronal responses were indeed related to reaction times in a multiple regression that controlled for other variables (Results, section ‘Reaction time control analysis’). However, these neuronal reaction time effects did not account for our main finding of sequence progress coding. We found no significant differences between basolateral and centromedial amygdala neurons with respect to relations to reaction times. We include an interpretation of these effects in the Discussion.

*3) Given the strong relationship between neural activity and progress on the choice task, it raises an interesting question about the activity on error trials. Do cells that show pre-fixation or fixation period progress-related activity increases/decreases exhibit aberrant activity on error trials? Or do other aspects of neural activity predict error trials?*

Thank you for this interesting suggestion. We have included a new analysis and result with respect to error trials (Results, section ‘Encoding of sequence progress by individual amygdala neurons’, third paragraph; Figure 3; Discussion, first and third paragraphs). Briefly, in a neuronal population analysis, the strength of sequence progress coding transiently declined on error trials and subsequently reappeared, which we interpret as additional evidence for the potential behavioral relevance of progress signals.

*4) What was the accuracy of decoding in the imperative control task? A similar figure as Figure 6 for comparison, along with a discussion of the implications of this analysis, would be useful.*

We have included the requested analysis in Figure 6 and in the Results section ‘Linear population decoding of sequence progress’, first paragraph and discuss the implications in the Discussion section, third paragraph.

*5) The observation of "adaptive encoding" is quite interesting. However, according to Figure 4, the adaptation seems only partial in that the normalized maximum firing rate is different for different sequence progress. Have the authors tested a partial adaption model, or adaptive models for reward magnitude and elapsed time (Figure 5)?*

Thank you for this interesting suggestion. We now include an additional analysis in which we tested for a partial adaptive model. The results are described in the Results, section ‘Adaptive model of sequence progress activity’, first paragraph; Eq. 8; and in Figure 5—figure supplement 1. Briefly, partial adaptation accounted better for our data than a non-adaptive model but across neurons was not significantly different from the full adaptive model. We did not test adaptive models for reward magnitude or elapsed time as we wished to restrict analysis to a comparison of model classes that were selected on theoretical grounds. For example, we did not consider an adaptive time model biologically plausible because this would presuppose that the animals know in advance the number of committed errors in a saving sequence. We provide an interpretation of the partial adaptive model in the Discussion, penultimate paragraph.

*6) From inspection of Figure 3, it appears that the majority of the sequence progress activity corresponded to a single, transient modulation that occurred at random times during a trial. The implications of this finding should be at least discussed.*

We now mention this observation in the Results, section ‘Encoding of sequence progress by individual amygdala neurons’, third paragraph, and discuss its implications in the Discussion, third paragraph.